# What Does It Take to Solve the 3D Ising Model? Minimal Necessary Conditions for a Valid Solution

**DOI:** 10.3390/e24111665

**Published:** 2022-11-15

**Authors:** Gandhimohan M. Viswanathan, Marco Aurelio G. Portillo, Ernesto P. Raposo, Marcos G. E. da Luz

**Affiliations:** 1Department of Physics, Federal University of Rio Grande do Norte, Natal 59078-970, RN, Brazil; 2National Institute of Science and Technology of Complex Systems, Federal University of Rio Grande do Norte, Natal 59078-970, RN, Brazil; 3Departmento de Física, Universidade Federal do Paraná, Curitiba 81531-980, PR, Brazil; 4Laboratório de Física Teórica e Computacional, Departamento de Física, Universidade Federal de Pernambuco, Recife 50670-901, PE, Brazil

**Keywords:** 3D Ising model, exactly solvable models

## Abstract

An exact solution of the Ising model on the simple cubic lattice is one of the long-standing open problems in rigorous statistical mechanics. Indeed, it is generally believed that settling it would constitute a methodological breakthrough, fomenting great prospects for further application, similarly to what happened when Lars Onsager solved the two-dimensional model eighty years ago. Hence, there have been many attempts to find analytic expressions for the exact partition function *Z*, but all such attempts have failed due to unavoidable conceptual or mathematical obstructions. Given the importance of this simple yet paradigmatic model, here we set out clear-cut criteria for any claimed exact expression for *Z* to be minimally plausible. Specifically, we present six necessary—but not sufficient—conditions that *Z* must satisfy. These criteria will allow very quick plausibility checks of future claims. As illustrative examples, we discuss previous mistaken “solutions”, unveiling their shortcomings.

## 1. Introduction

Exact results are always welcome in science, even if they are simplified or idealized models of more realistic natural phenomena [1]. For example, an elucidating discussion about the general importance of analytical solutions in physics can be found in Ref. [2]. The Ising model of magnetism was originally proposed by Wilhelm Lenz in 1920 and exactly solved in one dimension by his graduate student Ernst Ising in 1924. A summary of the thesis was published in 1925 [3]. The Ising model long ago ceased to be a paradigm restricted only to magnetism models. Currently, it has found applications in diverse areas, from neuroscience to sea ice and voter models, to name a few [4]. Since Lars Onsager’s solution of the 2D Ising model in 1942 (which was published in 1944 [5]), statistical physics in general and equilibrium statistical mechanics in particular have experienced a great flourishing of powerful mathematical techniques [6,7], allowing considerable progress towards obtaining exact expressions for many relevant models [8,9,10].

In spite of all these advances, one of the most paradigmatic systems in statistical physics, namely, the ferromagnetic Ising model with interactions between nearest-neighbor two-state spins on the simple cubic lattice [11]—henceforth referred as the 3D Ising model—has withstood all attempts at exact solution. According to Rowlinson [12], the first claim of analytically cracking the 3D Ising model was presented at StatPhys 2, held in Paris in 1952, by John R. Maddox, who later became editor of *Nature* (see also Refs. [13,14,15]). Eliott W. Montroll, still during the conference, showed that the proposed expression could not be correct, by comparing it with the first few terms of the well established exact series expansions for the high and low temperature limit cases. It was identified later that the error was due to an incorrect application of the Jordan–Wigner transformation. Since then, new announcements of an exact solution have been made every few years, only to be systematically proved incorrect (e.g., Refs. [16,17,18,19]). What is seemingly lacking, in this historical context, is a set of clear-cut plausibility criteria that can be used to quickly verify whether or not a claimed solution is minimally worth considering. Therefore, we shall list a number of necessary—but not sufficient—conditions that a correct partition function *Z* must satisfy. Although some of these have already (in part or in full) appeared in the literature, here they are presented in an unified, rigorous, and comprehensive way. We emphasize that on the one hand these conditions can be used to refute unfounded claims—and thereby to identify any inappropriate protocols that are responsible for the incorrect features for *Z*. On the other hand, they might serve as a guide for the development of promising solution schemes. Finally, we briefly mention two reasons why the 3D Ising model is considered a major open problem, besides the obvious intellectually instigating fact that it still stands eighty years since Onsager solved the 2D version. First, despite the simple definitions of Ising-like models, their usefulness in studying a large number of diverse processes is overwhelming [11,20,21,22,23,24,25,26,27]. However, this should not be a surprise. Indeed, given the universality classes of classical spin models, of which the Ising is probably the most emblematic example, they all can be mapped to distinct instances of the logic problem of satisfiability (SAT) [28]. In this way, the analytic solution of the 3D case conceivably would represent a great boost for its already wide applicability. Second, the 3D Ising model near the critical point is very closely related to string and gauge field theories [29]. In particular, these associations can be analyzed through conformal bootstrap methods [30] (for a review, see, e.g., [31]). Hence, the eventual determination (or eventual disproof of existence) of an exact solution certainly will impact other important areas of physics.

All these potential perspectives involving a rigorous analytical expression for the partition function of the 3D Ising model certainly justify the establishment of a minimal set of clear-cut criteria for the validation of its exact solution, as presented below.

## 2. The 3D Ising Model

Suppose a finite simple cubic lattice $Z_{N}^{3}$ of size L×L×L, with *L* a finite positive integer. There are $N=L^{3}$ sites that can be tagged as $l=(l_{1},l_{2},l_{3})$ for i=1,2,3 indicating the *i*th spatial direction (with unit vector ${\hat{e}}_{i}$) and $l_{i}=1,\ldots ,L$. Consider also two disjoint lattices, $\tilde{Z_{N}^{3}}$ and $\partial Z_{N}^{3}$, of sites labeled, respectively, by $l_{b}=\left(n_{1},n_{2},n_{3}\right)$, with $2\leq n_{i}\leq L-1$, and $l_{f}=\left(m_{1},m_{2},m_{3}\right)$, where at least one $m_{i}$ is necessarily 1 or *L*. Note that $Z_{N}^{3}=\tilde{Z_{N}^{3}}\cup \partial Z_{N}^{3}$, for $\tilde{Z_{N}^{3}}$ ($\partial Z_{N}^{3}$) representing the “bulk” (“frontier” or boundary) sites of $Z_{N}^{3}$. In other words, $\partial Z_{N}^{3}$ is the border or surface, whereas $\tilde{Z_{N}^{3}}$ is the inside region or interior of the lattice.

The spin variable $\sigma_{l}$ at the vertex $l\in Z_{N}^{3}$ can assume only two possible values, namely, −1 and +1. The Hamiltonian of the 3D anisotropic Ising model on $Z_{N}^{3}$ and with zero external magnetic field is given by
(1)$$\begin{array}{ccc} H_{N}\left(\sigma \right) & = & -\sum_{i=1}^{3}J_{i}\;\left(\sum_{l_{b}^{\prime \prime }-l_{b}^{\prime }={\hat{e}}_{i}}\sigma_{l_{b}^{\prime \prime }}\;\sigma_{l_{b}^{\prime }}+\;\sum_{l-l_{f}=\varepsilon \left(l\right)\;{\hat{e}}_{i}}\sigma_{l}\;\sigma_{l_{f}}\right),\\ & = & H_{N}^{\left(b\right)}\left(\sigma \right)+H_{N}^{\left(f\right)}\left(\sigma \right)\;, \end{array}$$
where ε(l)=±1 if $l\in \tilde{Z_{N}^{3}}$ and ε(l)=+1 if $l\in \partial Z_{N}^{3}$. The quantities $J_{1,2,3}$ are the couplings constants in the three distinct spatial directions, i=1,2,3. The two terms in the last equality in Equation (Equation 1) can readily be identified as the energy contributions from the system bulk *b* and frontier *f* regions for a state σ of the system. In fact, each σ represents a possible distinct configuration of −1 s and +1 s along the sites of the whole lattice and characterizes a specific system state.

Different boundary conditions (BCs) can be imposed to the problem [32,33]. They essentially specify constraints on the spin configuration of the set ${\left\{\sigma \right\}}_{\partial Z_{N}^{3}}$ (see below).

The canonical partition function is conventionally defined as $Z_{N}\left(K_{1},K_{2},K_{3}\right)=\sum_{\sigma }\exp\left[\beta \;H_{N}\left(\sigma \right)\right]$, where $\beta ={\left(k_{B}\;T\right)}^{-1}$ and $K_{i}=\beta \;J_{i}$ is the *i*th reduced temperature. The sum is over all the possible spin configurations {σ} over $Z_{N}^{3}$ (observing the specified BCs). The partition function per site in the thermodynamic limit, the object of our interest here, is defined as
(2)$$Z\left(K_{1},K_{2},K_{3}\right)=\lim_{N\to \infty }{\left[Z_{N}\left(K_{1},K_{2},K_{3}\right)\right]}^{1/N}\;.$$

The challenge of finding an exact analytic expression for the above *Z* has been called the “holy grail of statistical mechanics” [34]. To this day, it remains one of the most important unsolved problems in statistical physics.

The above seemingly simple model has a notorious history of errors and controversies. In 1925, soon after arriving at the exact solution of the 1D ferromagnetic Ising model (with an external field), Ising himself erroneously extrapolated to 2D one of his 1D findings [3]. Specifically, he generalized incorrectly to 2D and 3D his (correct) result that there can be no spontaneous magnetization in 1D for positive *T*. The historical details can be found in Ref. [35]. Another controversy related to the Ising model concerns the critical exponents for the 2D model. Up until Onsager’s exact solution of the model, it was assumed that the critical exponents were given by the mean field approximation. Onsager’s solution allowed the mistake to be caught [36]. A few years later, when Maddox in 1952 presented his claim of a solution to the 3D model, he applied the Jordan–Wigner transformation to effectively try to map 2D planes of spin operators into fermionic creation and annihilation operators [15]. The corresponding trick works for the 2D Ising model because linear chains of spin operators can indeed be mapped into fermionic creation and annihilation operators. However, this trick does not easily generalize to 2D planes. There are 2D generalizations of the Jordan–Wigner transforation, but thus far none of these methods have been shown to be useful for solving the 3D Ising model. All claims of an exact solution for the 3D model have been shown to contain errors (see Section 4).

## 3. Necessary Conditions for a Valid Solution

Our main goal in this contribution is to establish a set of six necessary conditions that must be satisfied by any prospective of an 3D Ising model exact analytic expression for $Z(K_{1},K_{2},K_{3})$. Although the list consists of necessary conditions, these are not also sufficient conditions. In other words, any claimed *Z* may be wrong even if all necessary conditions are satisfied. However, if even just one condition is violated, the claimed expression for *Z* is certainly wrong.

Next, we present the noted conditions in a order somehow going from the more basic and fundamental to the more technical and abstract.

### 3.1. Condition 1

*In the thermodynamic limit the per-site partition function Z of the 3D Ising model must be independent of the boundary conditions* [15,37,38].

Surprisingly, the very broad reach of this requirement seems not to be properly appreciated as one should expect. Some recent erroneous claims of an exact *Z* even have assumed that certain special BCs could violate Condition 1 (see Section 4). Given such misunderstandings, below we present a very general and rigorous (although concise) proof that this indeed must be the case.

The thermodynamic limit represented by Equation (Equation 2) is known to be equivalently stated in terms of sequences of subgraphs $G_{k}$ of $Z^{3}$ (e.g., see [39,40]). There is a large relative freedom in choosing the structure of the successive $G_{k}$s, provided they satisfy three fundamental properties, known as van Hove’s assumptions [38].

Denoting the number of vertices of a finite lattice *G* as V(G), these assumptions are as follows:$\cup_{k}\;G_{k}=Z^{3}$,$G_{k}\subset G_{k+1}$,$\lim_{k\to \infty }V(\partial G_{k})/V(G_{k})=0$, for $\partial G_{k}$ representing the frontier of $G_{k}$, namely, $\partial G_{k}=\{l\in G_{k}\;|\;\exists \;j\notin G_{k},\;|l-j|=1\}$.

From the above, it is also possible to define $\tilde{G_{k}}=\left\{l\in G_{k}\;|\;l\notin \partial G_{k}\right\}$, which is the bulk or core graph associated to $G_{k}$. As a trivial example of a $G_{k}$ satisfying all the above characteristics, we note the aforementioned limited cubic lattice $Z_{N}^{3}$, where $k=N=L^{3}$.

Thus, $Z(K_{1},K_{2},K_{3})$ can be written more generally as
(3)$$Z\left(K_{1},K_{2},K_{3}\right)=c\;\lim_{k\to \infty }\;{\left[Z_{G_{k}}\left(K_{1},K_{2},K_{3}\right)\right]}^{\frac{1}{V(G_{k})}},$$
where
$$Z_{G_{k}}\left(K_{1},K_{2},K_{3}\right)=\sum_{\sigma \in {\left\{\sigma \right\}}_{G_{k}}}\exp\left[\beta \;H_{G_{k}}\left(\sigma \right)\right]$$
and $H_{G_{k}}\left(\sigma \right)$ is the natural extension of Equation (Equation 1) to $G_{k}$. Further, *c* is a constant of a purely topological origin. It may differ from 1 depending on the characteristics of the chosen sequence $G_{k}$. However, it should not alter the resulting physics associated to the obtained partition function. Therefore, *c* might be set equal to 1 for the sake of discussion.

An important aspect of the finite Ising model relates to the boundary conditions assumed for the $G_{k}$s. A rather general formulation for typical BCs relies on the following construction. Let Ω(G) represent all the possible combinations of the spin configurations on the vertices of the finite *G*, i.e., an element of Ω is denoted by σ and is a map σ:G→{−1,1}. Consider then $\partial G_{k}$ and a specific subset $\Gamma_{BC}^{k}\subset \Omega \left(\partial G_{k}\right)$. We say that $\Gamma_{BC}^{k}$ determines the BCs on the Ising model if the allowed spin configurations $\sigma_{BC}$ belong to
$$\Omega_{BC}\left(G_{k}\right)=\Omega \left(\tilde{G_{k}}\right)\times \Gamma_{BC}^{k}.$$

For instance, for $\Gamma_{BC}^{k}=\Omega \left(\partial G_{k}\right)$, we have the usual free BCs, namely, for any site in the frontier the spin value can assume both values, −1 and +1, without restrictions. On the other hand, for $G_{k}$ displaying torus (or periodic), cylindrical, Klein, twisted, screw, etc., topology, then the permitted configurations in $\Gamma_{BC}^{k}$ are established by proper pairwise mappings in the form $\sigma_{l_{f}^{\prime \prime }}↔\sigma_{l_{f}^{\prime }}$. Hence, the finite partition function with the BCs determined by $\Gamma_{BC}^{k}$ reads
(4)$$Z_{G_{k}}^{BC}\left(K_{1},K_{2},K_{3}\right)=\sum_{\sigma \in \Omega_{BC}\left(G_{k}\right)}\;\;\;\exp\left[\beta \left(H_{G_{k}}^{\left(b\right)}\left(\sigma \right)+H_{G_{k}}^{\left(f\right)}\left(\sigma \right)\right)\right].$$

We can now easily show that in the proper limit the partition function is independent on the BCs. We first observe that, for the free BCs,
(5)$$\lim_{k\to \infty }\frac{\ln[Z_{G_{k}}^{\mathrm{free}}]}{V(G_{k})}$$
exists (for a proof, see Ref. [38]). Let us denote the limit in Equation (Equation 5) as $\ln[Z\left(K_{1},K_{2},K_{3}\right)]$. Second, we determine a bound for $H_{G_{k}}^{\left(f\right)}\left(\sigma \right)$ in Equation (Equation 4). For any site $l_{f}$ in $\partial G_{k}$, the maximum number of $l\in G_{k}$ such that $|l-l_{f}|=1$ is 5 for the simple cubic lattice, then we have that $|H_{G_{k}}^{\left(f\right)}|<5\;V\left(\partial G_{k}\right)$. Third, by lemma 2.2.1 in [37]
(6)$$|\ln\left[Z_{G_{k}}^{BC}\right]-\ln\left[Z_{G_{k}}^{\mathrm{free}}\right]|\leq \beta \;\left|H_{G_{k}}^{(f)\;BC}-H_{G_{k}}^{(f)\;\mathrm{free}}\right|.$$

Finally, by dividing Equation (Equation 6) by $V(G_{k})$, considering the triangular inequality
$$|H_{G_{k}}^{(f)\;BC}-H_{G_{k}}^{(f)\;\mathrm{free}}|\leq 2\times \beta \;5\;V\left(\partial G_{k}\right)$$
and since the sequence $G_{k}$ is van Hove, then
(7)$$\lim_{k\to \infty }\frac{\ln[Z_{G_{k}}^{BC}]}{V(G_{k})}=\lim_{k\to \infty }\frac{\ln[Z_{G_{k}}^{\mathrm{free}}]}{V(G_{k})}.$$

Thus, we readily conclude that
(8)$$\lim_{k\to \infty }{\left(Z_{G_{k}}^{BC}\left(K_{1},K_{2},K_{3}\right)\right)}^{1/V(G_{k})}=Z\left(K_{1},K_{2},K_{3}\right)$$
is well defined and independent on the BCs.

### 3.2. Condition 2


*The claimed partition function per site of the 3D model $Z(K_{1},K_{2},K_{3})$ must reduce to Onsager’s solution whenever one of the three reduced temperatures vanishes.*


Indeed, suppose without loss of generality that $J_{3}=0$, so that $K_{3}=0$. Then, for $l_{l}$ indicating the sites which lie in the plane $x_{3}=l$ (whose set of spin configurations we represent by $\left\{\sigma^{l}\right\}$), the 3D Hamiltonian of size $N=L\times L\times L=L^{2}\times L$ can be written as ($\sigma_{j}\equiv 0$ if $j\notin G_{k}$)
(9)$$H_{L^{2}\times L}\left(\sigma \right)=-\sum_{l=1}^{L}\;\sum_{l_{l}}\;\sum_{i=1}^{2}J_{i}\;\sigma_{l_{l}}\;\sigma_{l_{l}+{\hat{e}}_{i}}=\sum_{l=1}^{L}H_{L^{2}}\left(\sigma^{l}\right),$$
with $H_{L^{2}}\left(\sigma^{l}\right)$ the energy associated to the plane $x_{3}=l$ for $\sigma^{l}$ a given distribution of spins in such plane. Note that the 3D Hamiltonian is now expressed as the sum of *L* independent and identical 2D Hamiltonians. In this case, the 3D partition function $Z_{N}=Z_{L^{3}}\left(K_{1},K_{2},0\right)=Z_{L^{2}\times L}$ factors as
(10)$$Z_{L^{2}\times L}=\sum_{\sigma^{1}}\ldots \sum_{\sigma^{L}}\;\prod_{l=1}^{L}\exp\left[-\beta \;H_{L^{2}}\left(\sigma^{l}\right)\right]={\left(Z_{L^{2}}\right)}^{L},$$
where $Z_{L^{2}}=Z_{L^{2}}\left(K_{1},K_{2}\right)$ is the finite 2D partition function, whose per-site thermodynamic limit $Z(K_{1},K_{2})$ is naturally defined as
(11)$$Z\left(K_{1},K_{2}\right)=\lim_{L\to \infty }{\left[Z_{L^{2}}\left(K_{1},K_{2}\right)\right]}^{1/L^{2}}\;.$$

Thus, from Equation (Equation 10), we get
(12)$$\begin{array}{ccc} Z(K_{1},K_{2},0) & = & \lim_{L\to \infty }Z_{L^{2}\times L}^{1/L^{3}}=\lim_{L\to \infty }{\left(Z_{L^{2}}^{L}\right)}^{1/L^{3}}\\ & = & \lim_{L\to \infty }Z_{L^{2}}^{1/L^{2}}=Z\left(K_{1},K_{2}\right). \end{array}$$

Thus, the 3D per-site partition function must reduce to Onsager’s solution when any of the three couplings is made to vanish.

### 3.3. Condition 3


*For the isotropic case, namely $K=K_{i}$ (i=1,2,3), any claimed per-site partition function must be analytic for $0\leq K<K_{c}$ [41] and for $K_{c}<K<\infty$ [42], where $K_{c}=0.221\;654\;626\ldots$ is the well known numerically estimated value of the critical temperature of the 3D Ising model.*


Of course, when there are interactions besides nearest-neighbor or when there is an applied magnetic field, the range of analyticity in *K* for the partition function can be distinct from the above one (see, for instance, Refs. [38,41,42]).

Note also that an exact expression for *Z* should lead to an exact formula for $K_{c}$. Thence, if one has derived a tentative exact *Z*, its analyticity should be relatively easy to test, e.g., for complex functions from the Cauchy–Riemann equations [43], and for real functions using standard techniques, as those described in [44]. We observe that the numerical value of the critical temperature is known with very high precision from Monte Carlo simulations and other numerical approaches [45,46,47,48].

### 3.4. Condition 4


*For the isotropic case, the series expansion in the high (low) temperature limit—K small (large)—of the claimed solution must perfectly match the already established series to all known orders (see, e.g., Refs. [38,41]).*


This is a direct consequence of the fact that in the domain of an analytic function, its series expansion around a fixed expansion point must be unique.

Let $v_{i}=\tanh\left[K_{i}\right]$ be the high temperature expansion variable. For the isotropic case, i.e., $v=v_{i}$ (i=1,2,3), we define
(13)$$Z\left(K\right)=2\;{\left(\cosh\left[K\right]\right)}^{3}\;Z_{\mathrm{high}}\left(v\right).$$

For details, see, for instance, Equation (Equation 9) in Ref. [49]. This high temperature formulation is obtained by performing the sum over all states, and then using (for $\sigma_{i}=\pm 1$)
$$\exp\left[K\sigma_{i}\sigma_{j}\right]=\cosh\left[K\right]+\sigma_{i}\sigma_{j}\sinh\left[K\right]=\cosh\left[K\right]\left(1+\sigma_{i}\sigma_{j}\tanh\left[K\right]\right)\;.$$

This identity justifies the choice of the high temperature variable v=tanh[K]. It is well known (see Ref. [39] for the 2D analog) that $Z_{\mathrm{high}}\left(v\right)$ is then given by a properly defined generating function for graphs of given length, all of whose nodes have even degrees and whose edges only connect nearest neighbors, in the limit of an infinite lattice. Such graphs have been variously called “closed graphs” or “multipolygons.”

The first few expansion terms of $Z_{\mathrm{high}}\left(v\right)$ have been rigorously determined [38] via finite lattice methods [50]. From Condition 3, Z(K) is analytic for $T>T_{c}$. Therefore, for large *T*, the function $Z_{\mathrm{high}}\left(v\right)$ is also analytic. In this way, the series expansion of $Z\left(K\right)/(2\;{\left(\cosh\left[K\right]\right)}^{3})$ must coincide with the noted known terms.

Similarly, by setting $u_{i}=\exp\left[-4K_{i}\right]$ as the low temperature variable, for the isotropic case we have $u_{i}=u$ (i=1,2,3). Then, one can write the partition function as
(14)$$Z\left(K\right)=u^{-3/4}\;Z_{\mathrm{low}}\left(u\right)\;.$$

See, for instance, Equation (1.5) in Ref. [40]. This low temperature formulation simply ennumerates the number of configurations with a given domain wall area. Indeed, the quantity $Z_{\mathrm{low}}$ is the generating function, in the limit of an infinite lattice, of closed surfaces of given area (see, e.g., Ref. [40]). The prefactor $u^{-3/4}$ is due to the ground state energy not being zero. If each of the three bonds per site in the ground state contribute an energy −βJ=−K, then the partition function per site for the ground state is $\exp\left[-3K\right]=u^{3/4}$.

The first few exact terms of the low temperature expansion of $Z_{\mathrm{low}}\left(u\right)$ have been calculated [50]. Therefore, similarly to the high temperature expansion, for $T<T_{c}$ low enough, any claimed exact solution should meet term-by-term the noted series.

The first few expansion terms of both $Z_{\mathrm{high}}\left(v\right)$ and $Z_{\mathrm{low}}\left(u\right)$ are known, thanks to the finite lattice method [50]. Ian Enting and Tom de Neef originally developed this innovative approach in the 1970s for generating series expansions, with applications to exact enumeration problems [51]. Since then, this powerful numerical method has led to significant advances. For example, Iwan Jensen used it to calculate the statistics of self-avoiding polygons on the square lattice [52]. Very recently, Nathan Clisby wrote an expository article concerning the method’s relevance to the enumerative combinatorics of lattice polymers [53]. It is arguably the most powerful algorithmic technique for obtaining exact series expansions to high order of models whose exact solution is not known, including, of course, the 3D Ising model.

Table 1 lists the first few non-zero terms of the high and low temperature expansions. Therefore, according to the present condition, any claimed *Z* that does not exactly comply with the terms in Table 1 cannot be the exact partition function for the 3D Ising model.

#### 3.4.1. Condition 5


*The claimed exact Z should display permutation symmetry and convexity on the reduced temperature variables $K_{i}=\beta \;J_{i}$.*


The first property, namely permutation symmetry, as proposed in [15], implies that $Z\left(K_{1},K_{2},K_{3}\right)=Z\left(K_{\pi (1)},K_{\pi (2)},K_{\pi (3)}\right)$, where π(i) is a permutation for i=1,2,3. Since the final *Z* does not depend on the BCs or on the specific choice of the van Hove sequence, consider $G_{k}$ in Equation (Equation 3) as the simple cubic lattice $Z_{N}^{3}$ with free BCs. Then, the proof is direct since the finite $Z_{N}\left(K_{1},K_{2},K_{3}\right)$ trivially presents the aforementioned symmetry. Further (refer to Ref. [48]), if one can use *Z* to derive an equation for the critical $\beta_{c}$, i.e., $F(\beta_{c}\;J_{1},\beta_{c}\;J_{2},\beta_{c}\;J_{3})=0$, then also $\beta_{c}$ must be invariant under any permutation of $J_{1}$, $J_{2}$, and $J_{3}$.

In Ref. [38], it has been shown that, for the isotropic case $J_{i}=J$ (i=1,2,3), the partition function must be a convex function in β. For the anisotropic case, the proof follows exactly the same steps. Indeed, considering again $G_{k}$ the finite cubic lattice with free BCs, one finds that $G_{k}$ is convex (details omitted here). However, because the limit of a van Hove sequence of convex functions is also convex [54], the final result holds. As a consequence, one has the following. For α∈[0,1], $\beta_{a},\beta_{b}\in \left(0,\infty \right)$ and arbitrary $J_{1},J_{2},J_{3}$, then for $\beta_{d}=\alpha \;\beta_{a}+\left(1-\alpha \right)\;\beta_{b}$
(15)$$\begin{array}{ccc} Z(\beta_{d}\;J_{1},\beta_{d}\;J_{2},\beta_{d}\;J_{3}) & \leq & \alpha \;Z(\beta_{a}\;J_{1},\beta_{a}\;J_{2},\beta_{a}\;J_{3})\\ & & +\left(1-\alpha \right)\;Z\left(\beta_{b}\;J_{1},\beta_{b}\;J_{2},\beta_{b}\;J_{3}\right). \end{array}$$

#### 3.4.2. Condition 6


*The claimed exact solution must bring clarity to the conundrum related to the behavior of the partially resummed high temperature expansion of the anisotropic partition function [49,55].*


Based on the anomalous behavior of the partially resummed series solution of the 3D Ising model, it is now believed by many (but not all) that the 3D Ising model might “not be solvable” [55]. Here, by not solvable we mean that *Z* may not be a differentiably finite function (known in the mathematical jargon as a D-finite or holonomic function). Recall that $F(z_{1},z_{2},\ldots ,z_{N})$ is said holonomic if *F* is analytic in all the variables and satisfies a linear differential equation, whose coefficients are polynomials in $z_{1},z_{2},\ldots ,z_{N}$ (see, e.g., [56]). Hence, any claimed exact holonomic *Z* must be able to explain how and why the anisotropic 3D Ising model has a high-temperature series that, upon partial resummation, seems to indicate non-D-finiteness.

## 4. Some Previously Claimed Solutions in the Literature


We briefly review how previous advanced solutions have failed to satisfy the above conditions, thus not representing the correct exact $Z(K_{1},K_{2},K_{3})$.

As already noted, the proposal by Maddox in 1952 violated Condition 4 for the series expansions, as did the ones by Das [16], Lou and Wu [17], and Z.-D. Zhang [18]. The serious errors in the latter were also extensively addressed in Refs. [15,57]. Moreover, except for the solution proposed by Zhang, all others also violate Condition 3. In fact, Zhang’s solution only seems to satisfy it because the numerical value of the critical temperature is imposed as an ansatz, built into his construction. Still, Zhang’s critical temperature of $K_{c}\approx$ 0.221 658 637 208 698 [18] (taken from a conjecture of Rosengren [58]) differs from the best known numerical estimate $K_{c}\approx$ 0.221 654 626. It may seem that the discrepancy is small, but an exact solution should give the exact critical temperature to arbitrary precision, i.e., to all decimal places. For the exact critical temperature, it is not acceptable to tolerate a discrepancy in the 10th or even $10^{10}$th decimal place.

Crucially, none of the above claimed solutions minimally attend Condition 6. As explained above, the resummed high temperature series of the anisotropic 3D model seems to show an anomalous behavior, strongly suggesting non-D-finiteness. However, the claimed solutions all behave normally under resummation of the anisotropic high temperature series.

In 2021, D. Zhang [19] (not to be confused with Z.-D. Zhang) made another claim, promptly criticized in [48]. It is easy to check that the assertion in [19] violates Conditions 1 and 4. The claimed solution also fails to bring new insights regarding Condition 6. For Condition 3, the predicted critical temperature disagrees drastically with the known numerically estimated value. Finally, regarding Condition 2, Zhang writes (inaccurately) that:

When the interaction energy in the third dimension vanishes, Onsager’s exact solution of the 2D Ising model is recovered immediately. *This guarantees the correctness of the exact solution of the 3D Ising model* [emphasis added].

In fact, there is no such guarantee. Condition 2 is a necessary but not a sufficient condition for a solution to be correct. For example, the expression (for $v_{i}$ as defined before)
(16)$$\begin{array}{ccc} \ln[Z] & = & \ln[2\cosh\left[K_{1}\right]\cosh\left[K_{2}\right]\cosh\left[K_{3}\right]]\\ & & +\frac{1}{2\;{\left(2\;\pi \right)}^{3}}\;\int_{-\pi }^{\pi }dk_{1}\;dk_{2}\;dk_{3}\\ & & \times \ln[\left(1+v_{1}^{2}\right)\;\left(1+v_{2}^{2}\right)\;\left(1+v_{3}^{2}\right)\\ & & -2\;v_{1}\;\left(1-v_{2}^{2}\right)\;\left(1-v_{3}^{2}\right)\cos\left[k_{1}\right]\\ & & -2\;v_{2}\;\left(1-v_{1}^{2}\right)\;\left(1-v_{3}^{2}\right)\cos\left[k_{2}\right]\\ & & -2\;v_{3}\;\left(1-v_{2}^{2}\right)\;\left(1-v_{1}^{2}\right)\cos\left[k_{3}\right]], \end{array}$$
correctly reduces to Onsager’s solution of the 2D model if any one of the three $K_{i}$ are made to vanish. However, this expression clearly is not correct because it violates Conditions 1, 3, 4, and 6 above. See also the famous (but wrong) conjecture of Mark Kac, e.g., in Ref. [39].

## 5. Final Remarks and Conclusions

The last few decades have witnessed significant developments [59,60,61] aimed at obtaining an exact expression for the 3D Ising model. In the absence of strong theoretical results pointing otherwise, such steady progress should dispel the false myth regarding the (non)solvability of the 3D model (see below).

First, we emphasize that the ferromagnetic 3D Ising model with nearest neighbor interactions is *not* an NP-complete problem. Recall that a problem is said to be NP-complete when it can be used to simulate any problem classified as NP. A problem belongs to the NP complexity class when a proposed solution can be checked in polynomial time by a deterministic Turing machine. It is an open question to determine if NP problems can also be not just checked but also *solved* in polynomial time by a deterministic Turing machine (P complexity class). It is true that there is a theorem concerning NP-completeness of the Ising model due to Sorin Istrail [59]. Nonetheless, it refers to the 3D Ising spin glass with arbitrary interactions, not to the ferromagnetic model. Moreover, the problem addressed in [59] relates to finding the ground state. For the ferromagnetic case, the ground state is trivial, viz., with all spins aligned (so doubly degenerate).

Second, although there is strong numerical evidence of non-solvability of the 3D Ising model in terms of D-finite functions (see Condition 6), mathematical proofs for this supposition are still lacking. If indeed this would be the case, an exact analytic solution based on nonholonomic functions could still be possible. Actually, many experts have been careful to make clear that the aforementioned non-solvability of the 3D Ising model is conjectural. Barry Cipra, writing in *Science* [34], has stated that

It might still be possible to find exact answers for some special cases of the Ising model, Istrail notes. In particular, the ferromagnetic case of the 3D Ising model may turn out to be simple enough to solve.

And third, it is also not true that the progress is too slow or that the problem is hopelessly too difficult. Nor is it a waste of time—quite the contrary. In the preface to *Polygons, Polyominoes and Polycubes*, Anthony J. Guttmann writes [62],

This is indeed a golden age for studying such problems. With powerful computers and new algorithms, unimaginable numerical precision in our estimates of properties of many of these models is now possible. On the mathematical side, we are developing tools for solving increasingly complex functional equations, while the theory of conformal invariance, and the developments around stochastic Löwner evolution have given us powerful tools to predict, and in some cases to prove, new results. The scientific community in this field is divided into those who think we will never solve the problem, of say the perimeter or area generating function of self-avoiding polygons in two dimensions, and those who think that we will. I am firmly in the latter camp …

Finally, we briefly discuss other three dimensional lattice systems. Condition 1 is general and valid for all lattice systems, so long as the interactions in the Hamiltonian are nearest-neighbor. In contrast, conditions 2, 4, and 5 as formulated are specific to the simple cubic lattice. However, we can expect that there should be analogs of these conditions for each lattice system. The same should be true for Condition 3 and the numerical value of $K_{c}$. Condition 6, however, is the most difficult to generalize. Very little is known about the analog of Condition 6 for other lattice systems.

Summarizing, we have reviewed, systematized, and enlarged a set of necessary conditions characterizing a potentially exact *Z* for the 3D Ising model. We have arranged this set into a single framework. Obviously, this set does not per se establish a concrete protocol that can solve the Ising system. Nevertheless, if even a single criterion is violated, one can be 100% certain that the methodology followed is fatally flawed. In this sense, the advance reported here has the potential to guide the maturing of future attempts to obtain the true *Z*.

We emphasize that the discussion about misguided attempts in the literature presented here by no means has the intention of criticizing these authors. Our purpose is solely to illustrate the subtleties and intricacies of the problem, which has deceived even some of the most respectable researchers. Our discussion thus makes clear the real necessity of clear-cut tests to check the plausibility of future claims.

Finally, we note an eventual (although improbable) curious consequence of our results. An exact analytic expression for *Z* should observe *all* the previously addressed requirements. However, it could be the case that such a function—observing the full set of conditions 1–6—cannot exist. A proof, of course, would settle negatively the possibility of an analytical *Z*. However, we conjecture that the six conditions are not mutually inconsistent.

## Figures and Tables

**Table 1 entropy-24-01665-t001:** The first ten nonzero coefficients in the high temperature expansion of $Z_{\mathrm{high}}\left(v\right)=\sum_{n=0}^{\infty }a_{n}v^{n}$ and in the low temperature expansion of $Z_{\mathrm{low}}\left(u\right)=\sum_{n=0}^{\infty }b_{n}u^{n}$ (see main text). The coefficients were obtained by Guttmann and Enting using the finite lattice method [50].

*n* (High)	$a_{n}$	*n* (Low)	$b_{n}$
0	1	0	1
4	3	3	1
6	22	5	3
8	192	6	−3
10	2046	7	15
12	24,853	8	−30
14	329,334	9	101
16	4,649,601	10	−261
18	68,884,356	11	807
20	1,059,830,112	12	−2308

## Data Availability

Not applicable.

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
