# Peer review of "What Does It Take to Solve the 3D Ising Model? Minimal Necessary Conditions for a Valid Solution"

_entropy, 2022, doi:10.3390/e24111665_

Round 1

Reviewer 1 Report

Comments on: What does it take to solve the 3D Ising model? Minimal necessary conditions for a valid solution by G. M. Viswanathan and others.

The paper deals with the minimum set of necessary conditions for an exact partition function of the 3D Ising model. The model was proposed by Wilhelm Lenz, and first analyzed by his then student Ernst Ising, who presented an exact solution for the 1D case in his thesis.

          The paper brings new light to an old problem, as well as summarize several previous results in a simple and straightforward way.

Of the six necessary conditions presented in the manuscript, some of them are well known, like condition 2 ( “The claimed partition function per site of the 3D model must reduce to Onsager’s solution whenever one of the three reduced temperatures vanishes), while others are often overlooked, like condition 1 regarding independence, in the thermodynamic limit, of the per-site partition function independent on the boundary conditions.

          Section 4 in particular (“Some Previously Claimed Solutions in the Literature”) is very interesting, mainly when one remembers that the first false conclusions regarding the model was due to Ising himself, who extended his 1D results to two and three dimensions ( no phase transition at finite temperature of a two-state spins, which Onsager proved wrong some decades later). Also, the failed attempt of John Maddox, who presented his 3D solution at the StatPhys conference in Paris in 1952 is well known by everybody, but very few people actually know why the Maddox solution fails, and the present paper brings a digression on the matter.

I recommend the paper as it is, although it is always a good practice to entice the authors to check their manuscript for typos or grammar errors.

Author Response

Please see the attached PDF file

Reviewer 2 Report

Referee's report on

What does it take to solve the 3D Ising model? Minimal necessary conditions for a valid solution

by

G. M. Viswanathan et al.

The paper discusses a set of necessary conditions to be satisfied by a 3D Ising lattice such as to allow for a meaningful analytical solution in the thermodynamic limit. Altogether six “minimally necessary” conditions are formulated in Sec. 3, with a reference to the existing literature and ongoing debate in the field. For each single condition a comment (set of comments) is given illustrating its possible controversies and relevance for the problem. It is suggested that the paper reviews these conditions in a “systematized” (line 264) fashion justifying publication. Without disregarding this point of view completely I would like to note, however, that the conditions reviewed are such that each of them might constitute a problem per se, with its own mathematical formalism and literature background, so trying to merge them together in a brief outlook gives an impression (perhaps kind of erroneous) of several, apparently disparate pieces of work, loosely connected between each other. The presentation is then complicated by the fact that the authors touch upon the numerous controversies in the field, considered by different authors at different times, so for a common reader not familiar with the full history of the problem it is kind of uneasy to follow. In that regard, while the authors do give references to the various literature sources, the allowed details appear nevertheless sketchy, and the paper as whole, not self-consistent. I’m also kind of irritated by the use of very special terms fired off without explanation, for instance, what is non-D-finiteness in lines 193 and 207, or NP-completeness in line 230? More so, please explain in Sec. 3.0.4. where Eq. (13) comes from, and why the expansion goes over hyperbolic tangent of the temperature. Is it really so obvious? In a similar spirit, please explain the origin of the power-law pre-factor in Eq. (14) (I understood this form is just borrowed from other people’s works). What happens if the parameter u in Eq. (14) goes to zero? The temperature values in lines 203 and 204 seem to differ in the 5th order after comma, which as a comparison between analytical prediction (line 203) and numerical simulation (line 204) doesn’t sound too bad at the first glance (or am I missing something?), so what’s wrong about the tiny discrepancy here? I think, at least a few words of elucidation might be desirable. Concerning Condition 3, how could it be, otherwise, that the statistical weight is non-analytic outside the critical point? Further concerning the presentation, I’m not apologetic, in a research article, about a kind of ecstatic style, with multiple historical outlooks, details of biography of the different authors, even if the editors of Nature, further mixed with massive citations from their articles and details of controversies raised at the various conferences in the past, etc. I’d say all this could be relevant (and probably also interesting) during a conference chat or post-graduate lecture courses, but is hardly at the place in a research article. On top of that, I could not detect one single new result contained in this paper’s work (I have understood it was not even planned). The conclusions are kind of frustrating being merely reduced to a conjecture that the conditions reviewed could (hopefully) be not mutually inconsistent, the hint being that further research is in order (isn’t it like in every other field of active research?). In the preceding paragraph I read, in fact, that the purpose of the paper was just “to illustrate the subtleties and intricacies of the problem, which has deceived even some of the most respectable researchers.” Is it really enough, nowadays, for a research article?

I might recommend that the authors afford a critical revision of their work trying to produce a higher-quality output before it is considered for publication.

Author Response

Please see the attached PDF file.

Round 2

Reviewer 2 Report

I’m trustful that the authors have done their best in revising their paper in conformity with their style and previous agreement with the Editors. Given the extensive homework done in place I might recommend publication of this article in its revised form.